# Association between sarcopenia and osteoarthritis: A protocol for meta-analysis

Haochen Wang[☯], Ning Wang[☯], Yilun Wang, Hui Li[ORCID]*

Department of Orthopaedics, Xiangya Hospital, Central South University, Changsha, Hunan, China

☯ These authors contributed equally to this work.
* lihui1988@csu.edu.cn

**Data Availability Statement:** All relevant data are within the article and its Supporting information files.

**Funding:** H.L. is funded by the National Natural Science Foundation of China (81902265), and Y.W.

## Abstract

### Background

Sarcopenia, a relatively new syndrome referring to the age-related decline of muscle strength and degenerative loss of skeletal muscle mass and function, often resulting in frailty, disability, and mortality. Osteoarthritis, as a prevalent joint degenerative disease, is affecting over 250 million patients worldwide, and it is the fifth leading cause of disability. Despite the high prevalence of osteoarthritis, there are still lack of efficient treatment potions in clinics, partially due to the heterogeneous and complexity of osteoarthritis pathology. Previous studies revealed the association between sarcopenia and osteoarthritis, but the conclusions remain controversial and the prevalence of sarcopenia within osteoarthritis patients still needs to be elucidated. To identify the current evidence on the prevalence of sarcopenia and its association with osteoarthritis across studies, we performed this systematic review and meta-analysis that would help us to further confirm the association between these two diseases.

### Methods and analysis

Electronic sources including PubMed, Embase, and Web of Science will be searched systematically following appropriate strategies to identify relevant studies from inception up to 28 February 2022 with no language restriction. Two investigators will evaluate the preselected studies independently for inclusion, data extraction and quality assessment using a standardized protocol. Meta-analysis will be performed to pool the estimated effect using studies assessing an association between sarcopenia and osteoarthritis. Subgroup analyses will also be performed when data are sufficient. Heterogeneity and publication bias of included studies will be investigated.

### PROSPERO registration number

CRD42020155694.

is funded by the Youth Science Foundation of Xiangya Hospital (2021Q14). No funding bodies had any role in study design, data collection and analysis, decision to publish, or preparation of the manuscript.

## Introduction

With global population aging and increased longevity, aging and age-related diseases have become substantial burden and inevitable challenges worldwide. Sarcopenia, defined as an age-related muscle mass decline and muscle strength loss, results in reduced mobility, function and quality of life, and thus greatly increasing healthcare expenditures [1]. Although sarcopenia is a relatively new syndrome which was first described in the 1980s [2], it has become a common condition with an estimated prevalence from 12.9% to 40.4% with various diagnostic criteria [3, 4]. Sarcopenia is not only simply recognized as an age-related syndrome but also found to be correlated with increased risk of fall/fracture [5, 6], functional decline [7], multiple chronic diseases [8–10], loss of independence [11–13], frailty and mortality [14]. Sarcopenia is becoming a critical public health burden compounded by an expanding elderly population, being reported that the direct cost for medical spending due to sarcopenia was around $18.5 billion (i.e., 1.5% of the total health care spending) [15] for the year of 2000 in the United States, and since then, the economic burden of this progressive and generalized skeletal muscular disorder has grown substantially [16].

Osteoarthritis, the most common degenerative joint disease, is a leading contributor of physical disability nowadays [17, 18]. Since osteoarthritis has brought a severe impact on both individuals and the society as a whole, a comprehensive understanding of the underlying mechanism and potential risk factors of osteoarthritis has a significant importance [19]. Multiple types of risk factors have been identified to be correlated with pathogenesis of osteoarthritis [20], among which muscle weakness is considered as one of the major ones [21, 22]. For the various recommended intervention measures of osteoarthritis, functional exercise and muscle strength exercise have been drawing growing attention. Previous studies have suggested that there appears to be a bidirectional relationship between muscle weakness and osteoarthritis, muscle weakness might be a contributor to osteoarthritis progression and vice versa. On the one hand, as the atrophy or weakness of periarticular muscles would lead to the development, progression and severity of osteoarthritis, patients with osteoarthritis would adapt their lifestyle to sedentary and inactivity to avoid joint pain and stiffness [23–26]. Subsequently, sedentary and physical inactivity would in turn reduces energy expenditure and results in muscle wasting, thus would lower the joint-protective ability [27]. On the other hand, pain and stiffness of osteoarthritis joints cause physical inactivity, which would lead to adipose tissue gains and overweight development in these patients. The pressure of increased load further exacerbates the progression of osteoarthritis, and it is the combination of these factors that is considered to create and perpetuate a vicious cycle between muscle weakness and osteoarthritis [28, 29].

Yet, few studies considered muscle weakness or atrophy as a disease (i.e., sarcopenia) and the relationship between sarcopenia and osteoarthritis has remained ambiguous and no strong consensus has been reached [30]. Some suggested that sarcopenia was likely to positively correlate with osteoarthritis [31–34], and other studies did not support this observation [35, 36]. One of the plausible reason could be the definition of sarcopenia has been progressing and updating for decades, but full agreement on the involved variables and cutoff points has not reached yet [3], and this may lead to different prevalence rates. Furthermore, different anatomical location of osteoarthritis may exhibit different associations with sarcopenia. One study found that sarcopenia was associated with osteoarthritis at the hip and lower limbs [34], while another study reported that sarcopenia was independently associated with knee osteoarthritis and inversely associated with lumbar spine osteoarthritis [33]. One approach to synthesis existing knowledge is to identify consistencies across studies through a meta-analysis, but to our knowledge, no such study has systematically reviewed current evidence on the association between sarcopenia and osteoarthritis.

Therefore, this meta-analysis study aims to identify the association between sarcopenia and osteoarthritis more comprehensively. The results of this study will further our knowledge on whether sarcopenia and osteoarthritis are associated at different targeted joints, thereby enabling the development of preventive and therapeutic strategies for both sarcopenia and osteoarthritis.

## Methods

### Study design

This meta-analysis protocol has been registered with the international prospective register of systematic reviews PROSPERO network (registration number: CRD42020155694). The consent of this protocol is developed based on the Preferred Reporting Items for Systematic Review and Meta-Analyses Protocols (PRISMA-P) 2015 Statement Guidelines (S1 Appendix) [37].

### Eligibility criteria

The initially-retrieved studies will be evaluated for inclusion according to the following inclusion criteria: (1) observational studies including cohort studies, cross-sectional studies or case-control studies that focus on the prevalence of sarcopenia in patients with and without osteoarthritis, (2) diagnosis of sarcopenia using any definition criteria (e.g., low appendicular muscle mass criteria, or the European Working Group on Sarcopenia in Older People [EWGSOP] criteria including low handgrip strength and/or low walking speed in combination with low muscle mass), and (3) the age of included subjects are ≥60 years. Studies will be excluded if they are: (1) lack of reporting on study outcomes and (2) duplicate publications.

### Information sources

Three electronic databases (i.e., PubMed, Web of Science, and Embase) will be searched with appropriate search strategies from inception up to February 2022 from each platform or database. In addition, reference lists of the included literature and relevant systematic reviews will also be browsed to identify the eligible studies.

### Search strategy

The search will be carried out by combining keywords terms or medical subject heading terms (MESH) for eligible studies from the databases mentioned above. The same search terms will be adapted based on the specific requirements of different syntax rules. The electronic search strategy is listed in Tables 1–3.

### Study selection

Two investigators will screen the title and abstract of each retrieved study independently to identify eligible studies after removing duplicates. Full text will be reviewed according to the inclusion and exclusion criteria if the eligibility of studies is uncertain. Discussion will be made by consulting a third investigator for any disagreements between the two investigators. Studies will not be restricted on the language and publication date. Study selection will be documented and summarized based on the PRISMA flow diagram.

### Data extraction

After systematic literature search is carried out, two investigators will screen the the included studies independently and extract the following data from in a standardized format: name of

**Table 1. Draft of search strategy to be used using PubMed electronic database.**

| Number | Search terms |
|--------|--------------|
| 1 | "osteoarthritis"[Mesh] |
| 2 | osteoarthriti*[Title/Abstract] |
| 3 | osteoarthro*[Title/Abstract] |
| 4 | gonarthriti*[Title/Abstract] |
| 5 | coxarthriti*[Title/Abstract] |
| 6 | coxarthro*[Title/Abstract] |
| 7 | osteo?arthritis[Title/Abstract] |
| 8 | gonarthro*[Title/Abstract] |
| 9 | OR/1-8 |
| 10 | "sarcopenia"[Mesh] |
| 11 | "muscle weakness"[Mesh] |
| 12 | sarcopen*[Title/Abstract] |
| 13 | "muscle mass"[Title/Abstract] |
| 14 | "muscle volume"[Title/Abstract] |
| 15 | "muscle quality"[Title/Abstract] |
| 16 | "muscle size"[Title/Abstract] |
| 17 | "lean mass"[Title/Abstract] |
| 18 | "muscle strength"[Title/Abstract] |
| 19 | "grip strength"[Title/Abstract] |
| 20 | "gripping strength"[Title/Abstract] |
| 21 | "hand strength"[Title/Abstract] |
| 22 | "holding power"[Title/Abstract] |
| 23 | "grip dynamometer"[Title/Abstract] |
| 24 | handgrip[Title/Abstract] |
| 25 | "muscular atrophy"[Title/Abstract] |
| 26 | "muscle atrophy"[Title/Abstract] |
| 27 | "muscular dystrophy"[Title/Abstract] |
| 28 | "muscle dystrophy"[Title/Abstract] |
| 29 | "physical function"[Title/Abstract] |
| 30 | "muscle weakness"[Title/Abstract] |
| 31 | OR/10-30 |
| 32 | 9 AND 31 |

the author(s), year of publication, study design, setting of the study, data sources, study period, sample size, age range of the participants, sex distribution, prevalence of sarcopenia in patients with and without osteoarthritis. The effect sizes (i.e., odds ratio [OR], relative risk [RR] or hazard ratio [HR]) will be directly extracted or calculated on the basis of the relevant data in the original study as far as possible. If any data of interest is not available, we will contact the author(s) of the concerned study to obtain the supplemental data to the best extent. Any disagreements in data extraction will be consulting a third investigator to reach a consensus.

## Quality assessment

Two investigators will evaluated the quality of included studies independently according to the Newcastle-Ottawa Quality Scale (NOS) [38]. The NOS scale is a validated scale for non-randomized studies in meta-analysis that evaluates the risk of bias with broad perspectives: (1) the selection of the study groups; (2) the comparability of the groups; and (3) the ascertainment of

**Table 2. Draft of search strategy to be used using Embase electronic database.**

| Number | Search terms |
|---|---|
| 1 | 'osteoarthritis'/exp |
| 2 | osteoarthriti*:ti,ab,kw |
| 3 | osteoarthro*:ti,ab,kw |
| 4 | gonarthriti*:ti,ab,kw |
| 5 | gonarthro*:ti,ab,kw |
| 6 | coxarthriti*:ti,ab,kw |
| 7 | coxarthro*:ti,ab,kw |
| 8 | osteo*arthritis:ti,ab,kw |
| 9 | OR/1-8 |
| 10 | 'sarcopenia'/exp |
| 11 | 'muscle weakness'/exp |
| 12 | sarcopen*:ti,ab,kw |
| 13 | 'muscle mass':ti,ab,kw |
| 14 | 'muscle volume':ti,ab,kw |
| 15 | 'muscle quality':ti,ab,kw |
| 16 | 'muscle size':ti,ab,kw |
| 17 | 'lean mass':ti,ab,kw |
| 18 | 'muscle strength':ti,ab,kw |
| 19 | 'grip strength':ti,ab,kw |
| 20 | 'gripping strength':ti,ab,kw |
| 21 | 'hand strength':ti,ab,kw |
| 22 | 'holding power':ti,ab,kw |
| 23 | 'grip dynamometer':ti,ab,kw |
| 24 | handgrip:ti,ab,kw |
| 25 | 'muscular atrophy':ti,ab,kw |
| 26 | 'muscle atrophy':ti,ab,kw |
| 27 | 'muscular dystrophy':ti,ab,kw |
| 28 | 'muscle dystrophy':ti,ab,kw |
| 29 | 'physical function':ti,ab,kw |
| 30 | 'muscle weakness':ti,ab,kw |
| 31 | OR/10-30 |
| 32 | 9 AND 31 |

either the exposure or outcome of interest for case-control or prospective/ retrospective cohort studies, respectively [39]. For the cross-sectional studies, an adapted form of NOS will be used to evaluates the risk of bias [40, 41]. Studies with more than five stars will be considered as high methodological quality. In case of any discrepancies, a consensus will be reached through

**Table 3. Draft of search strategy to be used using Web of Science electronic database.**

| Number | Search terms |
|---|---|
| 1 | TS = (osteoarthriti* OR osteoarthro* OR gonarthriti* OR gonarthro*OR coxarthriti* OR coxarthro* OR osteo*arthritis) |
| 2 | TS = (sarcopen* OR "muscle weakness" OR "muscle atrophy" OR "muscle mass" OR "muscle volume" OR "muscle quality" OR "muscle size" OR "lean mass" OR "muscle strength" OR "grip strength" OR "gripping strength" OR "hand strength" OR "holding power" OR "grip dynamometer" OR handgrip OR "muscular atrophy" OR "muscular dystrophy" OR "muscle dystrophy" OR "physical function" OR "muscle weakness") |
| 3 | 1 AND 2 |

a discussion, with the assistance of a third reviewer when necessary. Studies with a high risk of bias (e.g., small-sample or low-quality studies) will be excluded and the reasons for their exclusion will be noted.

## Data analysis

All data will be statistical analyzed using the statistical software Review Manager 5.3 software. The study characteristics will be summarized in narrative texts and baseline tables. Specifically, effect sizes (the pooled OR, RR or HR) and corresponding 95% CIs will be calculated respectively. Statistical heterogeneity between the studies will be evaluated with $I^2$ values, for highly heterogeneous studies (>50%) a random-effects model will be used. A fixed-effects model will be applied to perform data pooling when the level of heterogeneity is not significant. The meta-analysis is set to a statistical significance as $p$ value < 0.05. When data are sufficient, this study will also perform subgroup analyses stratified by obesity (obesity and non-obesity) and different joints (hip, knee and hand).

## Assessment of publication bias

The publication bias among various studies will be assessed using the visual examination of funnel plot and Egger's test if ten or more studies are available. Asymmetric funnel plot may imply possible publication bias, small-study effects, or other factors. If asymmetry is caused by small-study effects, we will conduct sensitivity analysis by excluding these studies to explore how this affects the results and conclusions of the meta-analysis.

## Sensitivity analysis

Sensitivity analysis will be performed to test the robustness of pooled results regarding study characteristics and methodological quality by removing some of the small-sample or low-quality studies. If heterogeneity exists, sensitivity analysis will be re-run while removing poor quality data in a step-by-step wise.

## Discussion

As sarcopenia is a relatively new disorder with high incidence and prevalence in elderly population, it has seriously affected the health of the elderly throughout the world. It has been postulated that sarcopenia and osteoarthritis may be co-existing conditions [42]. But the pathophysiological mechanisms associated with sarcopenia and osteoarthritis are unclear. Plausible factors might include ageing, disuse and inflammation. Yet, the relevance of these findings has not been established. To explore the relationship between these two prevailing diseases, it is of great significance to conduct a meta-analysis to determine the impact of sarcopenia on osteoarthritis.

So far, there have been several studies on the correlation between sarcopenia and osteoarthritis. Of them, four studies suggested that sarcopenia was likely to positively correlate with osteoarthritis [31–34], and two studies showed that obesity and sarcopenic obesity, but not sarcopenia, were associated with osteoarthritis [35, 36]. In addition, one study found that sarcopenia was associated with osteoarthritis at the hip and lower limbs [34], while another study reported that sarcopenia was independently associated with knee osteoarthritis and inversely associated with lumbar spine osteoarthritis [33]. However, due to the variation in diagnostic criteria and classification of sarcopenia, the association between sarcopenia and osteoarthritis is still inconclusive [32, 35]. Previous studies analyzed sarcopenia by adopting different diagnosis standards and in relation to different weight-bearing joints osteoarthritis, which could be a possible reason why the literature findings were inconsistent. An earlier cross-sectional study

discussed the associations between low skeletal muscle mass and radiographic osteoarthritis of the hip, lumbar and knee joints, and the results showed that the skeletal muscle mass exhibited different associations with different joints [33]. Sarcopenia, as a disease affecting the whole body, may not only influence the knee joints but also other joints. According to the diagnostic criteria given by EWGSOP [1], sarcopenia can be diagnosed by the tests of muscle strength (usually based on grip strength), muscle mass (usually based on dual-energy X-ray absorptiometry or bioelectrical impedance analysis) and muscle function (usually based on gait speed, short physical performance battery, or time-up-and-go tests), which involve multiple joints of the body including hand, hip, and knee.

Nevertheless, sarcopenia, as well as sarcopenic obesity, have both been recognized as leading contributors of increased disability and mortality [14, 43]. Given that the effect of obesity towards osteoarthritis in previous studies, we will further perform subgroup and sensitivity analyses focusing on obesity or sarcopenic obesity as well. Previously, obesity, which is often represented by an increased body mass index (BMI) or body weight, was generally considered as a major risk factor of osteoarthritis [44]. However, in view of that the ratio of muscle mass over fat mass is changing constantly with the aging process [45–47], conventional anthropometric indicators, such as BMI and weight, may not be able to fully represent adiposity [48]. Recently, sarcopenia and sarcopenic obesity have been reported to be associated with a number of diseases including osteoarthritis [10, 29, 49]. Thus, subgroup and sensitivity analyses of sarcopenia, sarcopenic obesity and obesity will be conducted to better illustrate the relationship of these disorders.

The outcome of this meta-analysis may address an association between sarcopenia and osteoarthritis that is of pivotal importance to understanding the underlying mechanisms. The results from this study are also likely to inform healthcare a better decision-making treatment decision and to maximize the benefits of prevent and control osteoarthritis progression for limiting sarcopenia risk.

## Supporting information

**S1 Appendix.**
(DOCX)

## Acknowledgments

Everyone who contributed significantly to the work has been listed.

## Author Contributions

**Conceptualization:** Haochen Wang, Yilun Wang.

**Data curation:** Haochen Wang.

**Formal analysis:** Ning Wang.

**Methodology:** Haochen Wang.

**Writing – original draft:** Haochen Wang, Ning Wang.

**Writing – review & editing:** Yilun Wang, Hui Li.

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
