## [Decision Letter · Decision Letter 0]

13 May 2022

PONE-D-22-07545Prevalence of sarcopenia and its association with osteoarthritis: a protocol for meta-analysisPLOS ONE

Dear Dr. Wang,

Thank you for submitting your manuscript to PLOS ONE. After careful consideration, we feel that it has merit but does not fully meet PLOS ONE’s publication criteria as it currently stands. Therefore, we invite you to submit a revised version of the manuscript that addresses the points raised during the review process.

We look forward to receiving your revised manuscript.

Kind regards,

Osama Farouk

Academic Editor

PLOS ONE

Journal Requirements:

"This work was supported by the National Natural Science Foundation of China (81930071, 81902265, 82072502), and the Youth Science Foundation of Xiangya Hospital (2021Q14). No funding bodies had any role in study design, data collection and analysis, decision to publish, or preparation of the manuscript."

We note that you have provided funding information that is not currently declared in your Funding Statement. However, funding information should not appear in the Funding section or other areas of your manuscript. We will only publish funding information present in the Funding Statement section of the online submission form. 

"The funders had and will not have a role in study design, data collection and analysis, decision to publish, or preparation of the manuscript."

Reviewers' comments:

Reviewer's Responses to Questions

**Comments to the Author**

1. Does the manuscript provide a valid rationale for the proposed study, with clearly identified and justified research questions?

Reviewer #1: Yes

Reviewer #2: Yes

2. Is the protocol technically sound and planned in a manner that will lead to a meaningful outcome and allow testing the stated hypotheses?

Reviewer #1: Yes

Reviewer #2: Yes

3. Is the methodology feasible and described in sufficient detail to allow the work to be replicable?

Reviewer #1: Yes

Reviewer #2: Yes

4. Have the authors described where all data underlying the findings will be made available when the study is complete?

Reviewer #1: No

Reviewer #2: No

5. Is the manuscript presented in an intelligible fashion and written in standard English?

Reviewer #1: Yes

Reviewer #2: Yes

6. Review Comments to the Author

You may also provide optional suggestions and comments to authors that they might find helpful in planning their study.

Reviewer #1: Manuscript by Wang et al involves the description of protocol to find a relevance between sarcopenia and osteoarthritis. It could be a powerful tool to cumulate and show their relationship. Manuscript is well written. Protocol includes multiple databases which is beneficial and avoid publication biasness. Results of meta-analysis could be affected by quality of included studies. For example, there could be effect of small study (study using small sample size than study with large sample size) on the heterogeneity of results. Author should include the quality assessment for such studies. Format of references should be consistent.

Reviewer #2: Dear authors

It's a good idea of a significant problem of elderly with good high quality research design " sarcopenia and its association with osteoarthritis" Thank you for the successful scientific writing of the protocol. But I have some comments below:

General comments: The protocol is well written in a good scientific way, but a linguistic revision and rephrasing are needed to correct as:

- in Page8, line 42: " two investigators will be evaluate" replace by " two investigators will evaluate"

- page11, line 70: " To be correlate with" replace by " to be correlated".

The title:

"Prevalence of sarcopenia and its association with osteoarthritis", the aim of the study and the included studies are focusing on the association between osteoarthritis and sarcopenia, so delete the prevalence from the title. The prevalence is only could be estimated from the cross sectional study and not from the case control and cohort study.

Introduction:

At the end of the introduction "Therefore, this meta-analysis study aims to estimate the pooled prevalence of

sarcopenia and to identify its association with osteoarthritis more comprehensively".

The aim is not clear here and not matched with methods and the planned analysis, please concentrate on the association between the sarcopenia and osteoarthritis as you explained through all the protocol sections.

Revise the protocol from the title and afterwords to maintain the integrity between all sections.

Methods

In general all items were covered

in eligibility criteria: please justify the selected age (>60 years old)

Discussion:

On page 17, from line 210 and after: This is a section discussing few studies will be included in the studies , but it's better to include them with the other studies at the end of the study.

Include the subgroup analysis about the relation with adiposity and obesity in the methods section. Also the same regarding the association with osteoarthritis in different joints.

Best wishes

7. PLOS authors have the option to publish the peer review history of their article (what does this mean?). If published, this will include your full peer review and any attached files.

Reviewer #1: **Yes: **Geetika Aggarwal

Reviewer #2: No

---

## [Author Response · Author response to Decision Letter 0]

11 Jun 2022

Dear Editor, 

Thank you for the opportunity to submit a revision of our research paper, “Prevalence of sarcopenia and its association with osteoarthritis: a protocol for meta-analysis” [PONE-D-22-07545] for consideration for publication in PLOS ONE. 

We provide a point-by-point response to the Editors’ and Reviewers’ comments. We hope that our responses are satisfactory and that the changes we have made in the text reflect our responsiveness to the comments and suggestions. 

We look forward to your further assessment of this revised paper and thank you again for your consideration of our manuscript for publication in your journal. 

Yours sincerely, 

Hui Li, MD, PhD 

Department of Orthopaedics, Xiangya Hospital, Central South University 

lihui1988@csu.edu.cn

Academic Editor 

Comment 1: Please ensure that your manuscript meets PLOS ONE's style requirements, including those for file naming. The PLOS ONE style templates can be found at https://journals.plos.org/plosone/s/file?id=wjVg/PLOSOne_formatting_sample_main_body.pdf and https://journals.plos.org/plosone/s/file?id=ba62/PLOSOne_formatting_sample_title_authors_affiliations.pdf

Response: Per the editor’s comment, we have updated the format accordingly throughout the revised manuscript. 

Comment 2: We note that the grant information you provided in the ‘Funding Information’ and ‘Financial Disclosure’ sections do not match. 

Thank you for stating the following in the Funding Section of your manuscript:

"This work was supported by the National Natural Science Foundation of China (81930071, 81902265, 82072502), and the Youth Science Foundation of Xiangya Hospital (2021Q14). No funding bodies had any role in study design, data collection and analysis, decision to publish, or preparation of the manuscript."

We note that you have provided funding information that is not currently declared in your Funding Statement. However, funding information should not appear in the Funding section or other areas of your manuscript. We will only publish funding information present in the Funding Statement section of the online submission form.

"The funders had and will not have a role in study design, data collection and analysis, decision to publish, or preparation of the manuscript." 

Response: We are sorry for the confusion. We have updated the funding statement in cover letter and remove the funding information in manuscript. Please change the online submission on our behalf. 

Action: “Funding statement H.L. is funded by the National Natural Science Foundation of China (81902265), and Y.W. is funded by the Youth Science Foundation of Xiangya Hospital (2021Q14). No funding bodies had any role in study design, data collection and analysis, decision to publish, or preparation of the manuscript.” (Line 17-21 in Cover letter). 

Comment 3: In your Data Availability statement, you have not specified where the minimal data set underlying the results described in your manuscript can be found. PLOS defines a study's minimal data set as the underlying data used to reach the conclusions drawn in the manuscript and any additional data required to replicate the reported study findings in their entirety. All PLOS journals require that the minimal data set be made fully available. For more information about our data policy, please see http://journals.plos.org/plosone/s/data-availability. 

Response: Per the editor’s comment, we have updated the Data Availability statement in Cover letter. 

Action: “Data Availability statement All relevant data are within the article and its Supporting information files.” (Detailedly, retrieval and storage of study records, abstracts and full-text articles will be performed using EndNote 20. Subsequently, extracted data will be tabulated and upload as a Supporting information file.) (Line 23-27 in Cover letter)

Comment 4: PLOS requires an ORCID iD for the corresponding author in Editorial Manager on papers submitted after December 6th, 2016. Please ensure that you have an ORCID iD and that it is validated in Editorial Manager. To do this, go to ‘Update my Information’ (in the upper left-hand corner of the main menu), and click on the Fetch/Validate link next to the ORCID field. This will take you to the ORCID site and allow you to create a new iD or authenticate a pre-existing iD in Editorial Manager. Please see the following video for instructions on linking an ORCID iD to your Editorial Manager account: https://www.youtube.com/watch?v=_xcclfuvtxQ

Response: Per the editor’s comment, we have provided ORCID iD in the system. 

Comment 5: Please include captions for your Supporting Information files at the end of your manuscript, and update any in-text citations to match accordingly. Please see our Supporting Information guidelines for more information: http://journals.plos.org/plosone/s/supporting-information.

Response: Per the editor’s comment, we have updated the name of supporting information file and its in-text citation accordingly. 

Comment 6: Please review your reference list to ensure that it is complete and correct. If you have cited papers that have been retracted, please include the rationale for doing so in the manuscript text, or remove these references and replace them with relevant current references. Any changes to the reference list should be mentioned in the rebuttal letter that accompanies your revised manuscript. If you need to cite a retracted article, indicate the article’s retracted status in the References list and also include a citation and full reference for the retraction notice. 

Response: Per the editor’s comment, we have checked the reference list and confirmed it is complete and correct. All cited papers have not been retracted. 

 

Review #1: 

Comment 1: Manuscript by Wang et al involves the description of protocol to find a relevance between sarcopenia and osteoarthritis. It could be a powerful tool to cumulate and show their relationship. Manuscript is well written. Protocol includes multiple databases which is beneficial and avoid publication biasness. Results of meta-analysis could be affected by quality of included studies. For example, there could be effect of small study (study using small sample size than study with large sample size) on the heterogeneity of results. Author should include the quality assessment for such studies. Format of references should be consistent. 

Response: We appreciate the reviewer’s positive comments of our manuscript. 

First of all, quality assessment was conducted using Newcastle-Ottawa Quality Assessment Scale (NOS) and studies with an NOS score ≥ 5 were considered high quality. If there are low quality studies, data synthesis will be performed carefully. Sensitivity analyses excluding those studies with low quality will be conducted when necessary. 

Secondly, if there are ten or more studies in the meta-analysis, we will perform funnel plots and Egger’s regression test to investigate publication bias according to the Cochrane Handbook (REF: Page MJ, Higgins JPT, Sterne JAC. Chapter 13: Assessing risk of bias due to missing results in a synthesis. In: Higgins JPT, Thomas J, Chandler J, Cumpston M, Li T, Page MJ, Welch VA (editors). Cochrane Handbook for Systematic Reviews of Interventions version 6.3 (updated February 2022). Cochrane, 2022. Available from www.training.cochrane.org/handbook.). If asymmetric funnel plot is suggested by a visual assessment, we will perform exploratory analyses to investigate it and to investigate whether asymmetry is the result of publication bias, small-study effects, or other factors. If it is likely that asymmetry is caused by small-study effects, we will conduct sensitivity analysis by excluding these studies to explore how this affects the results and conclusions of the meta-analysis. 

Finally, we have checked the reference list and confirmed it is complete and correct. 

Action: “Studies with a high risk of bias (e.g., small-sample or low-quality studies) will be excluded and the reasons for their exclusion will be noted.” (Line 166-167) 

“Asymmetric funnel plot may imply possible publication bias, small-study effects, or other factors. If asymmetry is caused by small-study effects, we will conduct sensitivity analysis by excluding these studies to explore how this affects the results and conclusions of the meta-analysis.” (Line 184-187) 

Review #2: 

Comment 1: The protocol is well written in a good scientific way, but a linguistic revision and rephrasing are needed to correct as:

- in Page8, line 42: " two investigators will be evaluate" replace by " two investigators will evaluate"

- page11, line 70: " To be correlate with" replace by " to be correlated".

Response: We appreciate the reviewer’s comments and have corrected these errors accordingly. 

Action: “Two investigators will evaluate the preselected studies independently for inclusion, data extraction and quality assessment using a standardized protocol.” (Line 34-36) 

“Multiple types of risk factors have been identified to be correlated with pathogenesis of osteoarthritis [20],…” (Line 62-63)

Comment 2: The title:

"Prevalence of sarcopenia and its association with osteoarthritis", the aim of the study and the included studies are focusing on the association between osteoarthritis and sarcopenia, so delete the prevalence from the title. The prevalence is only could be estimated from the cross-sectional study and not from the case control and cohort study. 

Response: We appreciate the reviewer’s comments and have changed the title accordingly. 

Action: “Association between sarcopenia and osteoarthritis: a protocol for meta-analysis” (Line 1)

Comment 3: Introduction:

At the end of the introduction "Therefore, this meta-analysis study aims to estimate the pooled prevalence of sarcopenia and to identify its association with osteoarthritis more comprehensively".

The aim is not clear here and not matched with methods and the planned analysis, please concentrate on the association between the sarcopenia and osteoarthritis as you explained through all the protocol sections.

Revise the protocol from the title and afterwords to maintain the integrity between all sections.

Response: We appreciate the reviewer’s comments and have revised the protocol accordingly. 

Action: “Association between sarcopenia and osteoarthritis: a protocol for meta-analysis” (Line 1) 

“Meta-analysis will be performed to pool the estimated effect using studies assessing an association between sarcopenia and osteoarthritis. Subgroup analyses will also be performed when data are sufficient.” (Line 36-38) 

“…, but to our knowledge, no such study has systematically reviewed current evidence on the association between sarcopenia and osteoarthritis.” (Line 94-95) 

“Therefore, this meta-analysis study aims to identify the association between sarcopenia and osteoarthritis more comprehensively.” (Line 97-98) 

Comment 4: Methods

In general all items were covered in eligibility criteria: please justify the selected age (>60 years old)

Response: We appreciate the reviewer’s comment. Sarcopenia is defined as age-related loss of skeletal muscle mass plus loss of muscle strength and/or reduced physical performance. And the age cutoff points were set as 60 or 65 years old, according to consensus of European Working Group on Sarcopenia in Older People (EWGSOP) and Asian Working Group for Sarcopenia (AWGS) [1, 2]. Although there are several mechanisms that may be involved in the onset and progression of sarcopenia, we would like to focus on the primary etiology (age-related sarcopenia). Furthermore, it has been shown that the incidence of osteoarthritis increases rapidly between the ages of 50 years and 75 years, which is unlikely to have biased our meta-analyses [3]. 

1． Cruz-Jentoft AJ, Baeyens JP, Bauer JM, Boirie Y, Cederholm T, Landi F, et al. Sarcopenia: European consensus on definition and diagnosis: Report of the European Working Group on Sarcopenia in Older People. Age Ageing. 2010;39(4):412-23. Epub 20100413. doi: 10.1093/ageing/afq034.

2． Chen LK, Woo J, Assantachai P, Auyeung TW, Chou MY, Iijima K, et al. Asian Working Group for Sarcopenia: 2019 Consensus Update on Sarcopenia Diagnosis and Treatment. J Am Med Dir Assoc. 2020;21(3):300-7 e2. Epub 20200204. doi: 10.1016/j.jamda.2019.12.012.

3． Hunter DJ, Bierma-Zeinstra S. Osteoarthritis. Lancet. 2019;393(10182):1745-59. doi: 10.1016/S0140-6736(19)30417-9.

Comment 5: Discussion:

On page 17, from line 210 and after: This is a section discussing few studies will be included in the studies, but it's better to include them with the other studies at the end of the study.

Response: We appreciate the reviewer’s comment. As requested, discussion was adjusted and rephrased. 

Action: So far, there have been several studies on the correlation between sarcopenia and osteoarthritis. Of them, four studies suggested that sarcopenia was likely to positively correlate with osteoarthritis [31-34], and two studies showed that obesity and sarcopenic obesity, but not sarcopenia, were associated with osteoarthritis [35, 36]. In addition, one study found that sarcopenia was associated with osteoarthritis at the hip and lower limbs [34], while another study reported that sarcopenia was independently associated with knee osteoarthritis and inversely associated with lumbar spine osteoarthritis [33]. However, due to the variation in diagnostic criteria and classification of sarcopenia, the association between sarcopenia and osteoarthritis is still inconclusive [32, 35]. Previous studies analyzed sarcopenia by adopting different diagnosis standards and in relation to different weight-bearing joints osteoarthritis, which could be a possible reason why the literature findings were inconsistent. An earlier cross-sectional study discussed the associations between low skeletal muscle mass and radiographic osteoarthritis of the hip, lumbar and knee joints, and the results showed that the skeletal muscle mass exhibited different associations with different joints [33]. Sarcopenia, as a disease affecting the whole body, may not only influence the knee joints but also other joints. According to the diagnostic criteria given by EWGSOP [1], sarcopenia can be diagnosed by the tests of muscle strength (usually based on grip strength), muscle mass (usually based on dual-energy X-ray absorptiometry or bioelectrical impedance analysis) and muscle function (usually based on gait speed, short physical performance battery, or time-up-and-go tests), which involve multiple joints of the body including hand, hip, and knee. (Line 206-227)

Comment 6: Include the subgroup analysis about the relation with adiposity and obesity in the methods section. Also the same regarding the association with osteoarthritis in different joints.

Response: We appreciate the reviewer’s insightful comment and have added the corresponding content in the revised manuscript. 

Action: “When data are sufficient, this study will also perform subgroup analyses stratified by obesity (sarcopenia, sarcopenic obesity and obesity) and different joints (hip, knee and hand).” (Line 177-179)

---

## [Decision Letter · Decision Letter 1]

18 Jul 2022

Association between sarcopenia and osteoarthritis: a protocol for meta-analysis

PONE-D-22-07545R1

Dear Dr. Li,

We’re pleased to inform you that your manuscript has been judged scientifically suitable for publication and will be formally accepted for publication once it meets all outstanding technical requirements.

Kind regards,

Osama Farouk

Academic Editor

PLOS ONE

Additional Editor Comments (optional):

Reviewers' comments:

Reviewer's Responses to Questions

**Comments to the Author**

1. Does the manuscript provide a valid rationale for the proposed study, with clearly identified and justified research questions?

Reviewer #1: Yes

Reviewer #2: Yes

2. Is the protocol technically sound and planned in a manner that will lead to a meaningful outcome and allow testing the stated hypotheses?

Reviewer #1: Yes

Reviewer #2: Yes

3. Is the methodology feasible and described in sufficient detail to allow the work to be replicable?

Reviewer #1: Yes

Reviewer #2: Yes

4. Have the authors described where all data underlying the findings will be made available when the study is complete?

Reviewer #1: Yes

Reviewer #2: Yes

5. Is the manuscript presented in an intelligible fashion and written in standard English?

Reviewer #1: Yes

Reviewer #2: Yes

6. Review Comments to the Author

You may also provide optional suggestions and comments to authors that they might find helpful in planning their study.

Reviewer #1: The authors have fully addressed my concerns. After assessment of responses on academic editor and other reviewer comments, the authors replied well to majority of the comments. I think the revised protocol manuscript is ready for publication.

Reviewer #2: Dear authors

Thank you for the done revisions, I think it's now ready for publication.

I wish it will be an interesting study.

7. PLOS authors have the option to publish the peer review history of their article (what does this mean?). If published, this will include your full peer review and any attached files.

Reviewer #1: No

Reviewer #2: No

---

## [Editor Report · Acceptance letter]

26 Jul 2022

PONE-D-22-07545R1 

Association between sarcopenia and osteoarthritis: a protocol for meta-analysis 

Dear Dr. Li:

I'm pleased to inform you that your manuscript has been deemed suitable for publication in PLOS ONE. Congratulations! Your manuscript is now with our production department. 

Kind regards, 

on behalf of

Dr. Osama Farouk 

Academic Editor

PLOS ONE